# Optimization of Ultrasonic-Assisted Extraction and Purification of Zeaxanthin and Lutein in Corn Gluten Meal

**DOI:** 10.3390/molecules24162994

**Published:** 2019-08-18

**Authors:** Litao Wang, Weihang Lu, Jiali Li, Jinxia Hu, Ruifang Ding, Mei Lv, Qibao Wang

**Affiliations:** School of Pharmacy, Jining Medical University, Rizhao 276800, China

**Keywords:** zeaxanthin, lutein, corn gluten meal, ultrasonic-assisted extraction, purification

## Abstract

Zeaxanthin and lutein have a wide range of pharmacological applications. In this study, we conducted systematic experimental research to optimize antioxidant extraction based on detection, extraction, process amplification, and purification. An ultrasonic-assisted method was used to extract zeaxanthin and lutein with high efficiency from corn gluten meal. Firstly, the effects of solid-liquid ratio, extraction temperature, and ultrasonic extraction time on the extraction of zeaxanthin were investigated in single-factor experiments. The optimization extraction parameters of zeaxanthin and lutein with ethanol solvent were obtained using the response surface methodology (RSM) as follows: liquid–solid ratio of 7.9:1, extraction temperature of 56 °C, and extraction time of 45 min. The total content of zeaxanthin and lutein was 0.501%. The optimum extraction experimental parameters were verified by process amplification, and we confirmed that the parameters of the extraction process optimized using the RSM design are reliable and precise. Zeaxanthin and lutein from crude extract of corn gluten were separated and purified using silica gel column chromatography with the purity of zeaxanthin increasing from 0.28% to 31.5% (about 110 times) and lutein from 0.25% to 16.3% (about 65 times), which could be used for large-scale industrial production of carotenoids.

## 1. Introduction

Natural bioactive substances in animals and plants are important sources of clinical and health foods [1,2,3,4]. Zeaxanthin and lutein, with strong antioxidant activity, can significantly alleviate visual fatigue and reduce the risk of macular degeneration and cataracts [5,6,7,8,9]. They also function in regulating the animal immune response, enhancing macrophage activity, inhibiting the proliferation of tumor cells, and modifying cell surface markers and signal molecules [10,11]. Therefore, zeaxanthin and lutein are excellent food nutritional additives, and have been widely used in several fields, such as food, medicine, and cosmetics. Many countries have approved zeaxanthin and lutein from naturally-derived sources as food colorants and dietary supplements [12].

Corn is one of the most widely grown cereal crops and an important source of food for humans and livestock. Corn gluten meal, a by-product after processing into starch, contains many important carotenoids, such as zeaxanthin (1) and lutein (2), as shown in Scheme 1, respectively [13]. To ensure the full use of biological activated substances, many extraction techniques have been used for extracting carotenoids, including organic solvents, microwave-assisted methods, ultrasonic-wave-assisted methods, and supercritical fluid [14,15,16]. Among them, ultrasonic-wave-assisted extraction requires less time and is highly efficient, which may be due to the high cavitation effect of ultrasonic waves, and, more importantly, provides a high extraction rate under low temperature conditions, which is a new technology that has attracted widespread attention in the extraction of natural products [17,18,19]. Low temperature extraction conditions prevent the instability of zeaxanthin and lutein that occurs at high temperatures, thus improving the extraction rate. Therefore, ultrasound-assisted methods are a suitable choice for the extraction zeaxanthin and lutein from corn gluten meal.

The response surface method (RSM) is a combination of statistical and mathematical methods, and is an effective tool for solving the relationship between random variables and system responses of complex systems using statistical comprehensive experimental techniques. RSM has been widely used in the development, improvement, and optimization of natural product extraction conditions [20,21,22,23]. Zeaxanthin and lutein are isomers of each other and have almost the same chemical structure, only different in the position of one C=C bond (Scheme 1). Therefore, separating and detecting zeaxanthin and lutein by HPLC is difficult because of their similar polarities. Many researchers have attempted to separate and analyze zeaxanthin and lutein. Although good separation has been obtained on C30 and chiral columns, the results of the separation analysis were not satisfactory on the C18 column [24,25,26,27]. So, establishing a simultaneous detection method of zeaxanthin and lutein on the C18 column is important.

At present, the feasibility of using multi-response optimization of the extraction procedure via the RSM method has not yet been explored, and no systematic study has been conducted on the extraction and purification of zeaxanthin and lutein from corn processing by-products. Hence, we aimed to apply the RSM method to find the optimal extraction process parameters of zeaxanthin and lutein in corn gluten meal, and then the crude extract product was further purified to obtain high purity zeaxanthin and lutein. A rapid and simple HPLC method was established for the identification and quantification of zeaxanthin and lutein using the C18 column. These findings lay the foundation for the development of highly biologically active products from corn by-products.

## 2. Results and Discussion

### 2.1. Zeaxanthin and Lutein Method Development

Zeaxanthin and lutein are carotenoids, which are the isomeric compounds. The C18 chromatographic column has a weak ability to separate them, so it has a strict requirement for the selection and proportion of the mobile phase. In this study, a variety of solvents (methanol, acetonitrile, water, dichloromethane, and ethyl acetate) were used to screen mobile phase systems. The experimental results when using different solvents as the mobile phase showed that the methanol/water and acetonitrile/water systems are accompanied by a slight tailing of the zeaxanthin and lutein peaks during the separation process. After replacing the water with methylene chloride or ethyl acetate, we found that the peak shapes of zeaxanthin and lutein significantly improved, and the improvement effect of acetonitrile/methylene chloride was better due to the strong elution ability of acetonitrile and the better solubility of zeaxanthin and lutein by methylene chloride. Therefore, acetonitrile and dichloromethane were selected as the mobile phase system, and the chromatographic conditions were: acetonitrile/dichloromethane = 95:5 (*v*/*v*), detection wavelength 450 nm, flow rate 1 mL/min, and temperature 25 °C. Zeaxanthin and lutein were separated at baseline, the retention time was moderate, and the peak shape was sharp and symmetrical. Zeaxanthin and lutein almost reached the separation effect on the C30 column using acetone/water as the mobile phase [24,27], but they were not separated on the C18 column [27]. The extraction program, linearity range, precision, stability, reproducibility, and recovery rate were confirmed by the experiment. The peaks of zeaxanthin and lutein appeared at 11.4 min and 12.4 min with a separation factor of 1.07 and resolution of 1.27. The zeaxanthin and lutein retention factors are 5.73 and 6.16, respectively. The standard chromatogram is shown in Figure 1A, and the theoretical plate of the chromatographic column was not less than 3000.

#### 2.1.1. Precision and Stability Experiments

The precision of the method was determined by measuring zeaxanthin and lutein standard solutions in parallel five times. The results of the precision and stability experiments are shown in Table 1. The coefficient of variation (%RSD) values of the precision experiments were 1.44% and 1.77%, indicating that the test method had satisfactory precision and is suitable for actual sample determination and analysis of zeaxanthin and lutein content. The zeaxanthin and lutein sample solutions were placed at 25 °C for 12 h and analyzed every two hours. The %RSD values of the stability experiments were 1.93% and 1.73%, indicating that the sample has good stability over 12 h.

#### 2.1.2. Reproducibility and Recovery of Zeaxanthin and Lutein

The reproducibility of the zeaxanthin and lutein separation method was determined. The process was repeated five times and the 0.91% and 0.75% RSD values show that the determination method has good repeatability (Table 1).

Zeaxanthin and lutein recovery content was obtained from the determination of five samples (repeated five times). The recovery of zeaxanthin was 0.228 mg after adding zeaxanthin (0.121 mg). The recovery of lutein was 0.166 mg after adding zeaxanthin (0.101 mg). The recovery rate was calculated as follows:(1)$$\mathrm{Recovery}\;\mathrm{rate}\;(\%)=\frac{\left(\mathrm{detected}\;\mathrm{value}-\mathrm{original}\;\mathrm{value}\right)}{\mathrm{added}\;\mathrm{value}}\times 100$$

The recovery rate of zeaxanthin was 100.16% and the RSD was 2.40%; the recovery rate of the lutein was 98.81% and the RSD was 2.59%, which indicate that the determination method had sufficiently satisfactory accuracy to be used for actual sample analysis of zeaxanthin and lutein.

The methodological verification proved that the analysis method is accurate and reliable. In the determination of the sample, zeaxanthin, lutein, and impurity peaks had a high degree of separation, indicating the method can be used for the rapid qualitative and quantitative analysis of zeaxanthin and lutein in the product.

#### 2.1.3. Determination of Sample Content

The sample solution was prepared according to the “Sample preparation” Section 3.2, and the zeaxanthin and lutein contents were determined according to the optimized chromatographic conditions. The chromatograms of zeaxanthin and lutein determination are shown in Figure 1B. According to the peak area and the standard working curve of zeaxanthin and lutein, the zeaxanthin and lutein contents in corn gluten meal were 229 μg/g and 166 μg/g, respectively.

### 2.2. Analysis of Single Factor Test Results

Given the consistency in the variation trend in the extraction rate of zeaxanthin and lutein, the content of zeaxanthin could be used as a reference index in single factor and RSM tests. Choosing the right solvent is crucial to improving extraction efficiency. Ethanol has the strongest cell-penetrating ability, so zeaxanthin and lutein can be rapidly extracted from corn gluten meal. Ethanol is also the least toxic to the human body. An ultrasound power of 250 W with 40 kHz, as the most conventional ultrasonic apparatus in the laboratory, was applied for the extraction optimization. Therefore, the effects of solid–liquid ratio, temperature, and ultrasonic extraction time on the content of zeaxanthin and lutein were investigated in detail, and the parameters of extraction process were optimized with corn gluten meal as the raw material and ethanol as the extraction solvent.

#### 2.2.1. Effect of Liquid-to-Solid Ratio on Extraction

A high extraction rate can be obtained by selecting an appropriate liquid-to-solid ratio during solvent extraction. Extraction solvent deficiency may result in low extraction rates (incomplete extraction), and the solvent volume must ensure complete immersion of the plant matrix material, but excessive solvent may result in lower extraction rates and solvent waste. The zeaxanthin contents with different liquid–solid ratios were evaluated using single factor analysis. The effect of liquid–solid ratio on zeaxanthin and lutein content is shown in Figure 2A. The liquid-to-solid ratio was set to 3:1, 5:1, 7:1, 9:1, and 11:1 at a temperature at 60 °C and ultrasonic extraction time of 10 min. When the liquid-solid ratio was 7:1, the zeaxanthin and lutein contents were nearly maximized compared with the liquid-solid ratios of 9:1. Therefore, the optimum range of the liquid-solid ratio in extraction was determined to range from 7:1 to 9:1.

#### 2.2.2. Effect of Temperature on Extraction

Temperature is an important factor affecting extraction. Generally, the diffusion and mass transfer can be accelerated at higher temperatures, which is advantageous for improving the extraction efficiency. However, excessively high temperatures destabilize the highly active compound. Therefore, choosing the right temperature is key to achieving a high extraction rate. The effect of temperature on the contents of zeaxanthin and lutein was further studied as depicted in Figure 2B. The extraction temperatures were set at 30, 40, 50, 60, and 70 °C. The contents of zeaxanthin and lutein increased and nearly reached a maximum at 60 °C under the optimal liquid–solid ratio of 7:1 and an ultrasonic extraction time of 10 min. The contents of zeaxanthin and lutein slightly reduced when the temperature was 70 °C. Based on this result, best temperature range of the RSM test was found to range from 50 to 70 °C.

#### 2.2.3. Effect of Time on Extraction

Extraction time is another important parameter in solvent extraction. At its core is the process through which the active components in the plant substrate are transferred to the extraction solvent via diffusion and permeation. The effect of extraction time on the extraction content is displayed in Figure 2C. Ultrasonic extraction time was set to 20, 30 40, 50, and 60 min at the optimal liquid–solid ratio of 7:1 and temperature of 60 °C, and the extraction yields of zeaxanthin and lutein were found to have no significant change after 40 min and tended to stabilize, which indicates that the appropriate extraction time is probably between 30 and 50 min, and the extraction time of the RSM test was selected to range from 30 min to 50 min.

### 2.3. Optimization of Extraction by RSM

Single factor test could not investigate the interaction between the factors, whereas RSM test has many advantages such as faster testing times and obtaining reliable data. The effects of liquid–solid ratio, extraction temperature, and extraction time on the content of zeaxanthin were further studied using RSM based on the box-Behnken design (BBD) to optimize the extraction conditions. Three factors (liquid–solid ratio, extraction temperature, and ultrasound time) and three levels (5:1, 7:1, and 9:1 solid-liquid ratio; 40, 50, and 60 °C extraction temperature; and 30, 40, and 50 min ultrasound time) were adopted to design the RSM experiments. All tests were conducted in random order, and the list of experimental groups and the obtained results are provided in Table 2.

Multivariate regression fitting of data was performed using Design Expert software (Version 8.0.6; Stat-Ease Inc., Minneapolis, MN, USA) to obtain a function of the zeaxanthin extraction rate: Y = −1.27 × 10^3^ + 106A + 22.1B + 18.7C − 6.98A^2^ − 0.131B^2^ − 0.0790C^2^ + 0.149AB − 0.224AC − 0.164BC, where Y is the extraction yield of zeaxanthin (µg/g), A is the liquid−solid ratio (mL/g), B is the extraction temperature (°C), and C is ultrasonic extraction time (min).

The ANOVA in Table 3 shows the *p*–value was <0.0001 and the model *F*-value was 1673, which implies the regression model and model are highly statistically significant, respectively. A *p*-value < 0.01 of the model terms indicates that the regression model is extremely significant. The lack of fit item of a *p*-value of 0.1712 and an associated *F*-value of 2.82 implies that lack of fit is not significant relative to the pure error. The adjusted correlation coefficient (Adj-*R*^2^) and predicted correlation coefficient (Pred-*R*^2^) were 0.9989 and 0.9947, respectively, indicating that the model fits well with high correlation between the measured and predicted data from the regression model. Therefore, this model can be used to analyze and predict the extraction process conditions of zeaxanthin.

Response surface analysis was used to determine the effect of independent variables on the average extraction rate of zeaxanthin. The ordinate and the abscissa represent the extraction yields and any two variables, respectively. The three-dimensional profiles indicate how any two variables affect the extraction yield. The effects of temperature, liquid-solid ratio, and time of extraction on the extraction yield are shown in Figure 3. Extraction yield gradually increased with temperature, liquid–solid ratio, and time at approximately 40 to 60 °C, 3:1 to 7:1, and 30 to 45 min, respectively. Further increases in these parameters led to a decrease in the extraction rate of zeaxanthin. The surfaces have obvious upper convex in Figure 3a,b and slight upper convex in Figure 3c with a maximum value at the center of the response surface, which confirm the rationality of the experimental models. Based on the multivariate regression fitting equation, the optimized extraction conditions were obtained: 7.89:1 liquid–solid ratio, 56.4 °C extraction temperature, and 45.16 min ultrasonic extraction time, resulting in a predicted extraction yield of 214 µg/g.

After the optimum scheme was determined, the verification tests were performed at the optimum conditions of 7.9:1 liquid–solid ratio, 56 °C extraction temperature, and 45 min ultrasonic extraction time. Three parallel experiments were performed using screening scheme to obtain an average zeaxanthin yield of 212 µg/g, which was close to the predicted value 214 µg/g. These results demonstrate that the model is adequate for predicting the optimization. Therefore, the optimized extraction scheme obtained by the RSM test can be used to determine the optimal extraction conditions for zeaxanthin and lutein. Under this condition, the average weight of the crude extract was 0.396 g from 5 g corn gluten meal (yield = 7.92%), the zeaxanthin and lutein contents in crude extract were 0.28% and 0.23% by HPLC, the total yield was 397 μg/g, and the total content of zeaxanthin and lutein was 0.51%.

### 2.4. Verification of Process Amplification

To verify the feasibility of the optimized extraction experimental parameters in the production process. We performed a sample extraction 100× magnification experiment three times using the optimum conditions (liquid–solid ratio, 7.9:1; temperature, 56 °C, and extraction time, 45 min), and the yields of zeaxanthin and lutein and crude extract were calculated. The average total content of zeaxanthin and lutein was 0.745%, and the average yield of crude product was 6.28%. The experimental results confirmed that the parameters of extraction process optimized using RSM design are reliable and precise.

### 2.5. Purification of Zeaxanthin and Lutein

Silica gel is a commonly used purification medium with abundant adsorption groups on its surface. It has excellent adsorption capacity and separation degree for zeaxanthin, lutein, and other pigments. The irreversible adsorption rate of zeaxanthin and lutein on silica gel is only 7.6%, which is a good material for the purification of zeaxanthin and lutein. Silica gel was used as a separation medium for the purification of the crude product, which was wet-packed into a chromatography column with petroleum ether. The crude product was dissolved in petroleum ether and loaded; the mobile phase was petroleum ether:ethyl acetate = 7:3 (*v*/*v*), The pigment belt solution was collected in stages, the composition of the eluent was examined by TLC, and the eluate of the zeaxanthin and lutein fractions were evaporated solvent in a vacuum and freeze-dried. The samples before and after purification were qualitatively and quantitatively analyzed. The content of zeaxanthin increased from 0.28% to 31.5% (about 110 times) and lutein increased from 0.25% to 16.3% (about 65 times) after a single purification by silica gel column chromatography. The purity of silica-gel-purified zeaxanthin was higher than that of macroporous adsorption resin for lycopene (30-fold increase) [28,29]. The HPLC chromatograms and ultraviolet visible scanning spectrum (200–800 nm) of the purified sample and standard sample are shown in Figure 4A,B; the absorption curves of purified sample and standard sample are basically the same. No impurity absorption peaks were observed in the ultraviolet region. High-resolution mass spectrometry characterization showed that the molecular weight of zeaxanthin and lutein obtained by separation and purification was 568.4289 (theoretical value 568.4280). These results show that silica gel column chromatography has a good effect on the purification of zeaxanthin and lutein crude products.

## 3. Materials and Methods

### 3.1. Samples and Chemicals Reagents

Zeaxanthin standard (batch number: KJ0618SA14, purity ≥ 85%) and lutein standard (batch number: F04N6M5261, purity ≥ 90%) were purchased from Yuanye Biotechnology Company Ltd. (Shanghai, China). Corn gluten meal was obtained from Geely Fish Bait Factory (Henan, China). Silica gel (200–300 mesh) was obtained from Qingdao Jiyida Silica Reagent Company Ltd. (Qingdao, China). Dichloromethane (HPLC grade), 95% ethanol, and ethyl acetate (analytical grade) were obtained from Kemiou Chemical Reagent Company Ltd. (Tianjin, China). Acetonitrile (HPLC grade) was purchased from Yuwang Industrial Company Ltd. (Shandong, China). Petroleum ether (analytical grade) was purchased from Sinopharm Chemical Reagent Company Ltd. (Shanghai, China). Methylene chloride (analytical grade) was obtained from Fuyu Fine Chemical Company Ltd. (Tianjin, China).

### 3.2. Sample Preparation

We accurately weighed 2.5 mg lutein and 1.0 mg zeaxanthin, which were placed in a 25 mL brown volumetric flask. After dissolving in a mobile phase, the volume was adjusted and shaken to produce a mixed reference solution, which was stored in the dark and protected from light.

We used a proper amount of dried corn gluten meal, ground it into a powder, passed it through a 30-mesh sieve as an extraction material, and stored it in the dark at room temperature.

### 3.3. Zeaxanthin and Lutein Extraction Protocol

The ultrasound apparatus with a power of 250 W and 40 kHz was purchased from Kunshan Ultrasonic Instruments Co., Ltd. (Kunshan, China). We added 1 g corn gluten meal and 7 mL 95% ethanol into a round-bottom flask, which was blended and placed in water for ultrasonic extraction for 40 min at 60 °C. The extraction was repeated twice, and we collected the filtrate. The precipitated protein was filtered, then transferred into a volumetric flask, and stored in low temperature without light. All samples were filtrated using a 0.45 μm filter membrane, and then determined by HPLC.

### 3.4. HPLC Identification and Quantification

The LC-10A HPLC (Shimadzu, Kyoto, Japan) was equipped with LC-10ATvp binary pump (Shimadzu, Kyoto, Japan), SIL-10A autosampler (Shimadzu, Kyoto, Japan), and SPD-M10Avp detector (Shimadzu, Kyoto, Japan). The instrument was controlled by a computer and the Lab Solutions (Shimadzu, Kyoto, Japan) chromatography workstation was used to analyze the data. To obtain better chromatographic peaks of zeaxanthin and lutein, various mobile phase systems were optimized, and the optimal chromatographic separation conditions were determined. The Shimadzu^®^ C18 column (4.6 mm × 250 mm, 5 μm; Kyoto, Japan) was used for the analysis of the zeaxanthin and lutein contents. The optimum chromatographic conditions for the flow rate, detection wavelength, column temperature, and the injection volume were selected as 1.0 mL/min, 450 nm, 25 °C, and 20 μL, respectively. The mobile phase consisted of acetonitrile/dichloromethane (95:5, *v*/*v*).

### 3.5. Linear Range and Standard Curve Determination

Zeaxanthin standard substance was precisely weighed (20.0 mg) and dissolved in a 100 mL brown volumetric flask with the mobile phase as the stock solution (contain zeaxanthin 170 μg/mL), and stored at −18 °C in a refrigerator in nitrogen away from light. The concentrations (35, 70, 105, 140, and 175 μg/mL) were produced from a stock solution using the mobile phase, and the standard curve was established from the content measurement by linear regression. The fitting equation is y = 42155x + 291397. The coefficient of determination (*R*^2^) value (0.999) revealed a good linearity for the selected range of zeaxanthin (35–175 μg/mL).

Lutein standard substance with a lutein content of 90% was precisely weighed (27.8 mg) and dissolved in 100 mL mobile phase in a brown volumetric flask as the stock solution (containing 250 μg/mL lutein), which was stored in a −18 °C refrigerator and sealed with nitrogen. The stock solution was then diluted to the experimental design concentrations (50, 100, 150, 200, and 250 μg/mL) using the mobile phase (95:5, *v*/*v*) to produce five experimental working points by the relationship between peak area and concentration. The standard working curve was determined by linear regression fitting, and the fitting equation is y = 34632x + 317906. The *R*^2^ value (0.998) showed a good linearity for the selected range of lutein (50–250 μg/mL).

### 3.6. Data Analysis

The experimental results were repeated at least three times unless otherwise noted. Design Expert 8.0 (DE, Stat-Ease, Inc., Minneapolis, MN, USA) was used for analysis of experimental data and obtaining the response models. All analyses were performed in triplicate (unless specified), and we report the mean to eradicate any discrepancies as the final test result.

## 4. Conclusions

In this study, a HPLC method with high sensitivity and good repeatability was established for the simultaneous separation and detection of zeaxanthin and lutein using a C18 column. This method can be used as a quality control method for the determination of zeaxanthin and lutein in feed, food additives, health care products, and other products. Through the RSM test, we determined the optimum extraction parameters as follows: extraction solvent 95% ethanol, ratio of liquid to material 7.89:1, extraction temperature 56.4 °C, and extraction time 45.16 min. Under these conditions, the total yield of zeaxanthin and lutein was 397 μg/g. The purities of zeaxanthin and lutein were 31.5% and 16.3%, respectively, after purification by silica gel column. This work lays a foundation for the comprehensive use of corn gluten meal resources and the development of zeaxanthin- and lutein-related products.

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
