# Peer review of "Optimization of Ultrasonic-Assisted Extraction and Purification of Zeaxanthin and Lutein in Corn Gluten Meal"

_molecules, 2019, doi:10.3390/molecules24162994_

Round 1

Reviewer 1 Report

The authors have done a good job modifying the manuscript and the type of statistical analysis to better use the data to optimize the process. I still hold onto some of my previous comments and would recommend the authors have a language expert take a look at the manuscript to correct several grammatical errors (I have listed some below). I recommend some more revisions to the manuscript to better adjust the study towards a general scientific audience who would be interested in the significance and the implications of this study. I recommend acceptance of the manuscript after the revisions have been made.

Here are my comments:::

In abstract - please change "pharmacological activities" to "pharmacological applications". This would ensure that a more general scientific audience can understand what you are saying here. Activities in different fields of chemistry have slightly different meaning.

Abstract Line 12 - please change "from the aspects of" to "to optimize antioxidant extraction based on"

Pg 1 Line 41 - what are (1) and (2) after zeaxanthin and lutein??

Pg 2 Line 67 - Please change "characteristics of" to "advantages such as"

Pg 2 Line 82 - Please remove "the" in "the isomers of...".

Section 2.1.1 does not belong in results and discussion and should be moved to materials and methods.

Pg 2 Line 99-101 - So, C18 column cannot separate them but is there other ways to separate them? Has literature used something else before? There are no citations here indicating that this part of study is actually important.

Pg 2 Line 102 - which experimental results show that? It surely is not in the figure. I believe that authors wanted to say that "they observed peak tailing". 

Pg 3 Line 140 - Now we are crossing into materials and methods-type information. Please remove this and just hold on to what solvent selection meant for analytical method development.

Pg 10 Lines 558-560 - grammatically incorrect.

Due to the errors associated with the different methods, please restrict data reporting to 3 significant digits - please change this throughout the article.

Reviewer 2 Report

This work describes ultrasonic assisted extraction and HPLC separation of lutein and zeaxanthin in corn gluten meal. I have previously reviewed this work. The authors have made improvements to the work, but must make additional changes.

Observations:

The part of the extraction by UAE has been significantly improved, but the part of chromatographic separation by HPLC must be described in greater depth. In the section on the development of the chromatographic method, a more exhaustive study should be made or described, giving values of HPLC parameters of peak resolution, selectivity, etc. for each chromatographic peak until reaching the definitive method of separation.

The authors use an isocratic method, but I consider that a gradient method would be much more effective for the separation of these two carotenoids.

Section 3.3.: Describe the extraction equipment, characteristics, brand, city, country, as well as the extraction conditions (power, cycle, etc.).

Authors should be careful with the formal aspects in scientific writing and with the journal's style format.

Line 101: do not capitalize “A”.

Line 152, 171, 271, ….: Put a separation between a number and ºC. Unify and apply to the entire document. The authors have not correctly applied these corrections.

Lines 154, 163, 345, 348, 552, etc.: Put a separation after and before “=”, “-“, “<”, “x”.Unify and apply to the entire document.

Line 175, 278, …..: Put “n” in italics. Unify and apply to the entire document.

Lines 149, 165, 177…: Use the journal format. Do not put in italics.

Table 2: Put separations before each “(“.

Lines 348-351, Table 3, etc.: Unify the format of “p-value”. Put “p” in italics. Unify and apply to the entire document.

Lines 448, 457, 516, etc. Capitalize each word according the format of the journal. Unify and apply to the entire document.

Line 527: put separations between the number and the units.

Line 547: Include city.

Line 557: Include brand, city and country of the software.

References: Check the format of the references, for example, the volume should be in italics.

Round 2

Reviewer 2 Report

The authors have made most of the changes, so it can be considered to publish in Molecules with minor corrections.

Observations:

Line 27: zeaxanthin; lutein.

Line 165: “n” in italics.

Lines 197 and 212: Put a full stop after 2 and 3 respectively.

Lines 247, 255, 281, etc.: Capitalize each word according the format of the journal. Unify and apply to the entire document.

This manuscript is a resubmission of an earlier submission. The following is a list of the peer review reports and author responses from that submission.

Round 1

Reviewer 1 Report

Dear Authors:

Major comments

1.      In this study, authors try to optimize the extraction conditions of zeaxanthin and lutein in corn gluten meal through the single factor test and orthogonal test. However, it is more suitable for using Response Surface Methods for extraction condition optimization. In statistics, response surface methodology (RSM) explores the relationships between several explanatory variables and one or more response variables. There were several important reports in this journal that use RSM to optimize the extraction conditions. For example, extraction of Paclitaxel from Taxus x media using Ionic liquids as adjuvants 1, extraction of six major constituents from Ligusticum chuanxiong rhizome2, extraction polysaccharides and antioxidants of Atratylodes macrocephala 3, and extraction of Acer Truncatum leaves for maximal phenolic yield4. Please use the RSM for statistical analysis.

2.      The Identification of compounds should provide the NMR or Mass profiles instead of UV spectrum. Please add the data for compound identification.

Minor comments

1.      Please redraw the table 2, It is a five repeat of test, mean value and RSD is enough for results of recovery of zeaxanthin and lutein. Do not show the raw data as a result.

2.      Figure 2 is not necessary, please move it to supporting information.

3.      Figure 1 and Figure 3 should be presented in one figure, please combine it.

1          Tan, Z. et al. Ultrasonic Assisted Extraction of Paclitaxel from Taxus x media Using Ionic Liquids as Adjuvants: Optimization of the Process by Response Surface Methodology. Molecules 22, doi:10.3390/molecules22091483 (2017).

2          Liu, J. L. et al. Optimization of high-pressure ultrasonic-assisted simultaneous extraction of six major constituents from Ligusticum chuanxiong rhizome using response surface methodology. Molecules 19, 1887-1911, doi:10.3390/molecules19021887 (2014).

3          Pu, J. B. et al. Multi-Optimization of Ultrasonic-Assisted Enzymatic Extraction of Atratylodes macrocephala Polysaccharides and Antioxidants Using Response Surface Methodology and Desirability Function Approach. Molecules 20, 22220-22235, doi:10.3390/molecules201219837 (2015).

4          Yang, L. et al. Response Surface Methodology Optimization of Ultrasonic-Assisted Extraction of Acer Truncatum Leaves for Maximal Phenolic Yield and Antioxidant Activity. Molecules 22, doi:10.3390/molecules22020232 (2017).

Reviewer 2 Report

The authors have done an interesting study on optimizing ultrasonic assisted extraction of selected carotenoids from corn gluten. However, the study significantly lacks any value especially in the way that the authors have presented it. The abstract starts all wrong. With "representative" carotenoids and having pharmacological "activities" (did they mean "applications"?). As I read through the manuscript, the language is significantly lacking and serious efforts should be made to improve the language throughout the manuscript. I sincerely recommend that authors take help of a language service since I know that universities have them. 

Then, introduction is lacking. Why choose corn gluten? how much of these carotenoids can be found in corn gluten? Is it higher or lower than current sources? maybe it is cheaper than current way? do we know? 

The end of introduction section is, basically, the one that throws this study away when the focus of the study is indicated as improvement of HPLC procedure to analyzer the carotenoids. That may be an interest in validating the HPLC method to confirm the results from the study but the focus should have been optimization of the process itself. 

The results and discussion starts with the HPLC process validation and there is no indication as to why the specific parameters were chosen. Why acetonitrile and dichloromethane as mobile phase system? Were other methods tested? how do we know that this method is in any way better than the existing HPLC method? I understand from the data that it has stable performance but that does not mean that it is better than existing methods. However, I do understand that the validation of the method helps us with questioning the trends. But the way that HPLC results are discussed elevates its significance which, I personally feel, is very insignificant.

The orthogonal design is interesting. Why not use response surface method? Others have used it. What extra did we learn from the orthogonal design that others did not see? There is no comparison with the several results published in the literature on ultrasonic assisted extraction of these carotenoids (may not be from corn gluten but still trends are interesting to compare). I do not understand "conspicuousness" in the ANOVA table 4. I understand that its a statistical variable mainly associated with surveys but how does it apply to this scientific experiment?

The use of silica gel for purification of the carotenoids from corn gluten was a nice surprise at the end. Totally unexpected and added as an after-thought. Again, no comparison with existing technologies in literature. I am sure that the methods were obtained from other references. 

Citing all the above comments, I would recommend that the manuscript be rejected and authors revise and resubmit the manuscript once it is in an acceptable fashion.

Reviewer 3 Report

This work describes only the UAE optimization and purification of zeaxanthin and lutein in corn gluten meal. I consider that the article is very simple and deals with the study of aspects well known to the scientific community such as the separation of carotenoids in C18-HPLC or the extraction of carotenoids by ultrasound. In line 54 authors says “The focus of this study was to establish a rapid and simple HPLC method for the identification and quantification of zeaxanthin and lutein on C18 column” but they only indicate a simple method of separation in isocratic by HPLC, without any study of validation or robustness of the method. . On the other hand the English and the expressions need to be corrected by an expert English reviewer.

For all these reasons I consider that the work does not have the quality to be published in Molecules. I believe that the authors after making different changes and improving the article, should publish it in other journals that fit better as "Foods" or "Processes".

Other observations:

Figure 2 is not necessary. The information is redundant, and should appear only in the text.

Table 2 is not necessary. The information is redundant, and should appear only in the text or in supplementary material.

Section 2.7.: With the development of the method, the authors should indicate the "optimal" extraction conditions that derive from the design.

Figure 5: Indicate the measurement wavelength of the chromatogram. Indicate it also in the text.

Section 3.2. Section 3 should be expanded, addressing aspects such as origin, conservation, etc.

Section 3.3.: The extraction equipment is not described (brand, model, power, etc.). I consider that it is done in an ultrasonic bath.

Section 3.5.: The authors attach great importance to the development of the chromatographic method, but do not present either validation or robustness of the method. On the contrary they only present a method in isocratic. Using a gradient method could achieve a much more efficient separation.

Put a separation between a number and ºC. Unify and apply to the entire document.